# Enterotoxin Gene Cluster and *selX* Are Associated with Atopic Dermatitis Severity—A Cross-Sectional Molecular Study of *Staphylococcus aureus* Superantigens

**DOI:** 10.3390/cells11233921

**Published:** 2022-12-03

**Authors:** Leszek Blicharz, Maciej Żochowski, Ksenia Szymanek-Majchrzak, Joanna Czuwara, Mohamad Goldust, Krzysztof Skowroński, Grażyna Młynarczyk, Małgorzata Olszewska, Zbigniew Samochocki, Lidia Rudnicka

**Affiliations:** 1Department of Dermatology, Medial University of Warsaw, 02-008 Warsaw, Poland; 2Department of Medical Microbiology, Medial University of Warsaw, 02-004 Warsaw, Poland; 3Department of Dermatology, Yale School of Medicine, Yale University, New Haven, CT 06519, USA; 4Applied Analytics and Predictive Modelling Group, 00-001 Warsaw, Poland

**Keywords:** atopic dermatitis, enterotoxin, enterotoxin-like, enterotoxin gene cluster, microbiome, *selx*, *Staphylococcus aureus*, superantigens

## Abstract

*Staphylococcus aureus* superantigens (SAgs) have been reported to aggravate atopic dermatitis. However, comprehensive analyses of these molecules in multiple microniches are lacking. The present study involved 50 adult patients with active atopic dermatitis. *S. aureus* was isolated from the lesional skin, nonlesional skin, and anterior nares. Multiplex-PCR was performed to identify genes encoding (1) *selX* (core genome); (2) *seg, selI, selM, selN, selO, selU* (enterotoxin gene cluster, EGC); and (3) *sea, seb, sec, sed, see, tstH* (classic SAgs encoded on other mobile genetic elements). The results were correlated to clinical parameters of the study group. *selx* and EGC were the most prevalent in all microniches. The number of SAg-encoding genes correlated between the anterior nares and nonlesional skin, and between the nonlesional and lesional skin. On lesional skin, the total number of SAg genes correlated with disease severity (total and objective SCORAD, intensity, erythema, edema/papulation, lichenification and dryness). Linear regression revealed that AD severity was predicted only by *selx* and EGC. This study revealed that *selX* and EGC are associated with atopic dermatitis severity. Anterior nares and nonlesional skin could be reservoirs of SAg-positive *S. aureus*. Restoring the physiological microbiome could reduce the SAg burden and alleviate syndromes of atopic dermatitis.

## 1. Introduction

Atopic dermatitis (AD) is a common dermatological disorder with a lifetime prevalence of up to 20% in developed countries [1]. The clinical picture of AD involves the presence of eczematous lesions, pruritus, and a frequent coexistence of atopic diseases of the respiratory tract [1,2]. AD is a heterogenous disease developing as a consequence of genetic predisposition and the influence of environmental factors. The primary pathogenetic hallmarks of AD involve disruption of the epidermal barrier and immune dysregulation with a prominent Th2-skewing [3,4]. Recently, microbial dysbiosis of the skin has been implicated in AD onset and severity [5].

The microbiota of patients with AD are dominated by *Staphylococcus aureus* [6]. This phenomenon triggers loss of cutaneous homeostasis expressed by aggravation of epithelial barrier damage and inflammation. These observations have been attributed to virulence factors of *S. aureus,* such as exogenous toxins, immune-modulatory factors, and exoenzymes [7]. Of those, literature data suggest T-cell superantigens (SAgs) to be particularly significant in AD [8,9].

SAgs constitute the largest family of *S. aureus* exogenous toxins whose molecular weight ranges from 19–30 kDa [7]. They are divided into three groups: staphylococcal enterotoxins (SE), staphylococcal enterotoxin-like superantigens (SEls), and toxic-shock syndrome toxin 1 (TSST-1) [7]. SEs involve six SAgs (SEA to SEE and SEG) showing emetic activity. SEl group is formed by 15 SAgs (SElH to SElX) with a similar structure to SEs, but without the emetic activity. TSST-1, originally named SEF, was reclassified as a distinct SAg due to its association with toxic shock syndrome and lack of emetic activity.

SAgs are primarily encoded on mobile genetic element [10]. Therefore, their expression varies among different strains of *S. aureus*. Nevertheless, the enterotoxin gene cluster (EGC) involves six SAgs (SEG, SElI, SElM, SElN, SElO, SElU) that occur in a set particularly often [11]. Furthermore, selected novel SEls were identified as core genome elements [12]. Of those, SElX seems to be a particularly potent SAg whose influence on AD has not been fully analyzed [13].

Studies involving a comprehensive assessment of different SAg groups in multiple microniches of patients with AD are lacking. Therefore, the aim was to analyze the prevalence of genes encoding *selX*, members of the EGC (*seg, selI, selM, selN, selO, selU)*, and classic staphylococcal superantigens (*sea, seb, sec, sed, see, tstH*) on lesional skin, nonlesional skin, and in the anterior nares of patients with AD and to correlate the results to selected clinical parameters of the study group.

## 2. Methods

Adult patients (>18 years old) with active AD diagnosed based on the Hanifin and Rajka criteria were involved [14]. Patients with other active skin disorders and with a history of immunosuppressive therapy were excluded. Sporadic, short-term use of antibiotics due to infection over two months before the examination was allowed. The routine treatment of AD including emollients, topical corticosteroids, topical calcineurin inhibitors, and oral antihistamines was withdrawn five days before the evaluation. Before enrollment, the patients provided an informed consent for participation and for the use of the isolated biological material in the future.

Clinical examination was performed by one investigator (LB). Disease severity was determined using the SCORAD index. Total SCORAD was semi-quantified to classify AD severity (<25 points—mild, 25–50 points—moderate, >50 points—severe). The maximum extent of skin lesions during AD flares in the last year and during stable periods of the disease was determined based on the Wallace rule of nines [15]. The impact of AD on the patients’ quality of life was assessed using dermatology life quality index (DLQI). Total IgE serum concentration was measured using the ELISA method (the UniCap Fluorometer).

### 2.1. Staphylococcus aureus Isolation

Swabs for microbiological examination were taken from the lesional skin, nonlesional skin, and the anterior nares. Swabs from lesional skin were collected from the most intense lesion. Nonlesional skin swabs were routinely taken from the volar forearm and, if that area was involved, from another non-inflamed region of the skin. Swabs were taken with cotton-wool-tipped swab sticks immersed in 0.85% NaCl solution (bioMérieux, Marcy-l’Étoile, France) and secured in a transport medium (MedLab). Nasal samples were taken by performing clockwise and anti-clockwise 360° turns in both nostrils, and skin samples by rubbing an area of 4 cm^2^ (2 cm × 2 cm) field of the skin for 5 s. Within 24 h the swabs were plated on mannitol-salt agar medium (bioMérieux) and incubated for 24 h under aerobic conditions at 37 °C. In case of insufficient growth, the incubation was prolonged up to 48 h. After incubation, all morphologically distinct colonies which caused the yellowing of the medium were isolated and re-inoculated to other plates with mannitol-salt agar using a reduction seeding technique so as to obtain a pure laboratory culture.

Identification of *S. aureus* was performed by means of the VITEK MS (bioMérieux) mass spectrometer based on MALDI-TOF (Matrix-Assisted Laser Desorption Ionization Time-of-Flight) technology.

All identified strains of *S. aureus* were subsequently frozen in the temperature of −70 °C in brain-heart infusion broth for further analyses. Only viable strains of *S. aureus* were included in the analysis of SAgs.

### 2.2. Genomic DNA Extraction

Bacterial genomic DNA was extracted using a commercial kit (Genomic Mini, A&A Biotechnology, Gdańsk, Poland). First, 1.5 mL of log growth phase bacterial culture cultivated in brain-heart infusion medium (BHI) was applied to Eppendorf probes and centrifugated for 3 min at 13,000 RPM. The supernatant was discarded, and the sediment was suspended in 100 μL of Tris buffer. During the next stage, 10 μL of lysostaphin (1 U/μL concentration, Sigma-Aldrich) was added and the probes were incubated at 37 °C for 30 min. Then, 200 μL of LT lysing solution and 20 μL of proteinase K were subsequently added, and the probes were incubated for 20 minutes at 37 °C followed by 5 min at 70 °C. After the incubation, the probes were intensively vortexed for 20 s and centrifugated for 3 min at 13,000 RPM. The obtained supernatant was transferred to the DNA purification minicolumns and centrifugated for 1 min at 13,000 RPM. After rinsing with A1 solution twice, the minicolumns were transferred to new sterile 1.5 mL Eppendorf probes. Next, 100 uL of Tris buffer heated up to 75 °C were applied to the deposit and incubated for 5 min at room temperature, after which the minicolumns were centrifugated at 13,000 RPM. Following centrifugation, the minicolumns were discarded, and the genomic DNA obtained in the Eppendorf tube was stored at −20 °C until further analysis.

### 2.3. Detection of S. aureus Superantigen-Coding Genes

Thirteen staphylococcal enterotoxin genes, i.e., *sea, seb, sec, sed, see, seg, selI, selM, selN, selO, selU, selX*, and *tstH* were detected using the PCR technique and adequate primer pairs described by Salgado-Pabon et al. and Vu et al. with certain modifications introduced in this study (changes described below) [16,17]. The names, sequences, and the origin of starters as well as the size of acquired PCR products are presented in Table 1. PCR reactions were conducted using one set of primers in the case of *tst*H and multiple sets of primers for other genes (multiplex PCR). The following variants were performed: (1) *tstH* alone; (2) *sea* and *sec*; (3) *seg, selI, selU*; (4) *selM, selO, selX;* and (5) *seb, sed, see, selN*. The conditions for PCR reactions were the same for all combinations except for different annealing temperatures, which were 52 °C, 52 °C, 54 °C, 50 °C, and 52 °C for variants 1–5, respectively.

### 2.4. Identification of the Obtained PCR Products

Amplification products were subjected to electrophoretic separation in 1% agarose gels stained with ethidium bromide (Sigma-Aldrich, St. Louis, MO, USA). Eight μL of DNA loading dye were added to every probe (6x DNA Loading Dye, Fermentas), and fifteen μL of the prepared mixture were applied to adequate wells in the electrophoretic gel. Electrophoretic separation (45 min at 120V, 400 mA) was performed with DNA ladder applied to one of the wells (Gene Ruler 100 bp DNA Ladder, Thermo Fisher Scientific, Waltham, MA, USA). The gels were analyzed in UV light (wavelength 324 nm) using the GelDoc XR+ system and Image Lab Software (Bio-Rad, Hercules, CA, USA).

### 2.5. Statistical Analysis

Frequency tables were used to describe qualitative variables, and a typical measure of position (mean, median) and variability (standard deviation) were used to describe quantitative variables. For selected pairs of variables, associations/correlations were examined. The Fisher test was used to test relationships between categorical variables. To compare two groups, the Wilcoxon rank sum test with continuity correction was used, except attributes with a normal distribution, where the Welch Two Sample t-test was applied. The Spearman rank correlation method was used to measure the degree of the relationship between quantitative and order variables. Multivariate linear regression was used to predict the effect of covariates on the dependent variables. To identify normal distribution, the Shapiro–Wilk normality test was used. For applied methods, the standard level of *p*-value threshold was assumed (*p* < 0.05).

## 3. Results

A total of 50 adult patients with AD were involved in the study (17 females, 33 males; mean age: 33 ± 12.9 years). Detailed clinical characteristics of the study group are shown in Table 2.

The rates of *S. aureus* carriage were 43/50 (86%) on lesional skin, 33/50 (66%) on nonlesional skin, and 37/50 (74%) in the anterior nares. The differences in *S. aureus* carriage were statistically significant between lesional and nonlesional skin (*p* = 0.008), but not between other combinations of the analyzed microniches (*p* > 0.05). There were no differences in the carriage of *S. aureus* in any of the analyzed locations between males and females (*p* > 0.05).

The total number of identified genes encoding SAgs was 76 on lesional skin (1.52 ± 2.1 per patient), 82 on nonlesional skin (1.64 ± 2.15 per patient), and 52 in the anterior nares (1.04 ± 1.511 per patient). There was a significant difference in the number of SAgs per patient between the anterior nares and nonlesional skin (*p* = 0.012), but not between any other combination of the analyzed microniches (*p* > 0.05). The number of patients with non-SAg producing *S. aureus* was 13/43 (30.2%) on lesional skin, 5/33 (15.1%) on nonlesional skin, and 11/37 (29.7%) in the anterior nares (*p* > 0.05). Figure 1 shows a visualization of the products of exemplary multiplex-PCR used to detect genes encoding SAgs.

To verify whether the number of SAg-encoding genes correlated between the analyzed localizations, implying the circulation of virulent *S. aureus* in patients with AD, the Spearman rank correlation test was used. There was a positive correlation in the number of SAgs between the lesional skin and nonlesional skin (rho = 0.427, *p* = 0.016) and between the anterior nares and nonlesional skin (rho = 0.435, *p* = 0.026), but not between the anterior nares and lesional skin (*p* > 0.05).

The following genes were identified most frequently:on the lesional skin: *selX* (23/43 strains, 53.5%), *selN* (9/43 strains, 20.9%), *seg, selI, selM, selO, selU* (all identified in 7/43 strains, 16.3%). *see* was not identified in any of the isolates;on the nonlesional skin: *selX* (22/33 strains, 66.7%), *selN* (10/33 strains, 30.3%), *seg, selI, selM, selO, selU* (all identified in 9/33 strains, 27,6%). *seb, sed,* and *see* were not identified in any of the isolates;in the anterior nares: *selX* (15/37 strains, 40.5%), *selN* (9/37 strains, 24.3%), *selM, selO* (both identified in 5/37 strains, 13.5%). *seb* and *see* were not identified in any of the isolates.

Detailed information on genes encoding SAgs detected in *S. aureus* strains isolated from the lesional skin, nonlesional skin, and the anterior nares of the study group is presented in Table 3.

To determine the effect of the carriage of SAg-encoding genes on the values of clinical parameters in AD patients, the Wilcoxon rank sum test with continuity correction or the Welch Two Sample t-test (in case of variables with normal distribution) was used.

The presence of *selx* on lesional skin was associated with higher mean values of objective SCORAD (44.5 ± 10.21 vs. 34.88 ± 17.25, *p* = 0.036), extent (46.74 ± 31.79% vs. 29.85 ± 30.57%, *p* < 0.05), intensity (10.04 ± 1.99 vs. 8.26 ± 3.79, *p* < 0.04), dryness (1.96 ± 0.56 vs. 1.15 ± 1.03, *p* < 0.004), and the largest extent of skin lesions during the flares in the year preceding the examination (70.65 ± 29.17% vs. 43.85 ± 34.53%, *p* < 0.004).

The analysis of EGC genes on lesional skin revealed the following associations:*selN*—higher mean values of total SCORAD (58.73 ± 14.6 vs. 47.04 ± 17.14, *p* < 0.03), intensity (11.33 ± 3.24 vs. 8.59 ± 3, *p* < 0.04), redness (2.22 ± 0.97 vs. 1.49 ± 0.68, *p* < 0.03), and lichenification (2.44 ± 0.53 vs. 1.80 ± 0.78, *p* < 0.03);*seg, selI*, and *selU*—higher mean values of total SCORAD (58.14 ± 16.53 vs. 47.68 ± 17.02, *p* < 0.05) and lichenification (2.57 ± 0.53 vs. 1.81 ± 0.76, *p* < 0.02);*selM* and *selO*—higher mean values of objective SCORAD (49.9 ± 13 vs. 37.58 ± 14.84, *p* < 0.05), intensity (11 ± 1.83 vs. 8.77 ± 3.27, *p* < 0.03), and dryness (2.29 ± 0.49 vs. 1.40 ± 0.93, *p* < 0.02)

The presence of *sec* on lesional skin was associated with higher mean values of total SCORAD (63.67 ± 3.21 vs. 48.22 ± 17.30, *p* < 0.0003;) and DLQI (15 ± 1 vs. 12.63 ± 5.56, *p* < 0.04).

The presence of *seg, selI*, and *selU* on nonlesional skin correlated with lower mean values of the extent of skin lesions (17.33 ± 18.45 vs. 42.07 ± 32.72%, *p* < 0.02). Higher skin dryness was associated with the presence of *selX* (1.95 ± 0.65 vs. 1.18 ± 0.98, *p* < 0.007), *selM* and *selO* (2.22 ± 0.67 vs. 1.37 ± 0.92, *p* < 0.02).

The presence of *seg, selI*, and *selU* in the anterior nares was associated with lower mean values of the extent of skin lesions in stable periods of AD (0.67 ± 1.15 vs. 4.94 ± 3.64, *p* < 0.05). Lower mean values of itch were identified in patients with *sea* (3 ± 1 vs. 6.19 ± 2.74, *p* < 0.05)*, selM*, and *selO* (3.6 ± 0.55 vs. 6.27 ± 2.79, *p* < 0.04 in both cases).

Next, we explored the cumulative effect of the number of identified SAgs on the clinical parameters using the Spearman rank correlation. There was a positive correlation between the total number of SAgs identified on the lesional skin and disease severity (semi-quantitative assessment of total SCORAD–rho = 0.51, *p* < 0.0002; total SCORAD–rho = 0.45, *p* < 0.002; objective SCORAD–rho = 0.45, *p* < 0.002; intensity–rho = 0.48, *p* < 0.0004; redness–rho = 0.29, *p* < 0.05; swelling–rho = 0.35, *p* < 0.02; lichenification–rho = 0.41, *p* < 0.003; and dryness–rho = 0.42, *p* < 0.003). In case of the nonlesional skin, the number of SAGs correlated with intensity (rho = 0.28, *p* < 0.05) and dryness (rho = 0.31, *p* < 0.03). Selected results of the Spearman correlation and binary analysis are presented in Figure 2.

Multiple linear regression was used to test if the carriage of *selX,* any genes encoded on the EGC, and any of the classic staphylococcal superantigens on lesional skin significantly predicted the markers of disease severity.

Considering all those variables, the overall regression model was statistically significant for total SCORAD (R^2^ = 0.159, F (3,46) = 4.88, *p* = 0.045), objective SCORAD (R^2^ = 0.205, F (3,46) = 4.88, *p* = 0.014), intensity (R^2^ = 0.229, F (3,46) = 4.88, *p* = 0.007), redness (R^2^ = 0.159, F (3,46) = 4.88, *p*= 0.045), lichenification (R^2^ = 0.167, F (3,46) = 4.88, *p* = 0.037), dryness (R^2^ = 0.241, F (3,46) = 4.88, *p* = 0.005), and the largest extent of lesions during flares in the year preceding the examination (R^2^ = 0.158, F (3,46) = 4.88, *p* = 0.046).

*selx* significantly predicted objective SCORAD (β = 9.59, *p* = 0.021), dryness (β = 0.8, *p* = 0.002), intensity (β = 1.803, *p* = 0.0037), and the largest extent of lesions during flares in the year preceding the examination (β = 27.87, *p* = 0.005).

EGC significantly predicted total SCORAD (β = 11.67, *p* = 0.048), objective SCORAD (β = 11.84, *p* = 0.02), intensity (β = 3.04, *p* = 0.048), redness (β = 0.7, *p* = 0.01), and lichenification (β = 0.67, *p* = 0.013).

No variable was significantly predicted by genes encoding classic staphylococcal SAgs.

## 4. Discussion

SAgs are powerful mitogens causing polyclonal proliferation of up to 50% of T cells by crosslinking the Vβ T-cell receptors with major histocompatibility class II molecules on antigen-presenting cells [7]. This process impairs targeted adaptive immunity against *S. aureus* and triggers exaggerated inflammatory reaction. Literature data suggest an increased severity of AD in patients colonized by SAg-encoding *S. aureus* [18,19]. SAgs were also associated with resistance to topical steroids and tacrolimus, aggravation of itch in an IL-31 dependent manner, and IgE-mediated sensitization [20,21,22,23]. However, compared with this study, most reports considered only *S. aureus* isolates from the lesional skin and analyzed a smaller number of SAgs.

The present study confirmed a universally reported tendency for hypercolonization of patients with AD by *S. aureus*. This phenomenon was observed in all the analyzed microniches (lesional skin, nonlesional skin, and anterior nares). The lesional skin was the most frequently colonized location, followed by the anterior nares and nonlesional skin. High rates of *S. aureus* carriage in the anterior nares may suggest their role as a reservoir of *S. aureus*, maintaining chronic microbial dysbiosis observed in AD. Adult males and females did not show differences in the rates of *S. aureus* carriage, underlying the probable lack of sex-dependent differences in antimicrobial defenses.

The mean number of genes encoding SAgs in *S. aureus* strains isolated from the lesional skin, nonlesional skin, and the anterior nares was lower than in studies assessing a comparable number of SAgs on the skin of patients with AD [20,24]. Of note, the number of SAgs produced by *S. aureus* isolated in different populations is highly variable due to the coding of most staphylococcal superantigens on mobile genomic elements [25].

Although characterized by the lowest rates of *S. aureus* carriage, the nonlesional skin was demonstrated to harbor the strains with the highest number of SAg-encoding genes. It could be hypothesized that the microenvironment of the nonlesional skin is more challenging than that of the anterior nares (natural expression of staphylococcal adhesin ligands) [26] and the lesional skin (profound skin barrier disruption leading to increased expression of staphylococcal adhesin ligands) [27], which selects *S. aureus* strains capable of secreting the largest number of toxins, facilitating their survival.

A total of 15.1% to 30.2% of *S. aureus* strains isolated from the lesional skin, nonlesional skin, and the anterior nares were not SAg gene carriers. These data are concurrent with those reported in other studies [18,28]. Together with high prevalence of *S. aureus* in the analyzed microniches, this observation implies a considerable risk of exposition to SAgs in patients with AD. There were no differences in the presence of particular SAgs in any of the analyzed microniches between adult males and females, which underlies the lack of *S. aureus*’s ability to adapt to colonizing members of different sexes based on the secreted SAgs.

The total number of genes encoding SAgs correlated between all analyzed microniches, apart from the anterior nares and lesional skin. At the same time, a positive correlation between the total number of SAg-encoding genes and disease severity expressed by SCORAD index and its components was discovered on lesional skin and nonlesional skin. These observations could imply an indirect transmission of SAg-producing *S. aureus* from the anterior nares to the nonlesional skin, and a subsequent cutaneous dissemination of virulent strains of this pathogen with resulting aggravation of disease severity.

The most common SAg-encoding gene present in *S. aureus* isolates from the lesional skin, nonlesional skin, and the anterior nares was *selX*, followed by the genes of the EGC (*seg, selI, selM, selN, selO,* and *selU*). Genes encoding classic staphylococcal superantigens, i.e., *sea-e* and *tstH* were only identified in single strains. This contrasts with other reports stating the high carriage of *S. aureus*, producing SEs and TSST-1 in patients with atopic dermatitis [18,20].

SEl-X is a novel SAg regarded to be an element of *S. aureus* core genome [12,29]. It is reportedly present in 85%–100% of all strains. However, in the present study, the detection rate of *selX* was dependent on location, reaching a maximum of 66.7% on nonlesional skin. This seems to imply that although SEl-X might play an important pathogenetic role in AD, it is not vital for the survival of *S. aureus* on the skin and mucous membranes and therefore might be present with variable frequency. These observations concur with other reports revealing both the possibility of lower (≥50%) likelihood of encoding *selX* in certain *S. aureus* genotypes, as well as smaller (0%–80%) rates of detection of the protein product on lesional and nonlesional skin of patients with AD [24,30]. Lower expression of the *selx* gene (76.4%) was also reported in a study of individuals with bacteriemia and healthy controls [31].

Initially, SEl-X was revealed as a major virulence factor expressed by USA300, a CA-MRSA strain of *S. aureus* responsible for the epidemic of skin and soft tissue infections, necrotizing pneumonia, and extreme pyrexia [32]. Importantly, USA300 has been reported to cause infection in otherwise healthy individuals. Its significance was first noted based on the observation of increased lethality in an animal model of necrotizing pneumonia [32], but another report demonstrated that this could also be dependent on the presence of other virulence factors expressing superantigen-like behavior [33]. Furthermore, a study of asymptomatic nasal carriers, patients with isolated bacteremia, and bacteremia with infective endocarditis-associated *selX* locus to invasiveness of *S. aureus* [34].

On the molecular level, SEl-X was shown to manipulate both innate and acquired immune response by causing the chaotic activation of a large subset of T cells and inhibiting the neutrophil function by disrupting IgG-mediated phagocytosis [35,36]. The latter was reported to depend on the binding of adhesion molecules including P-selectin, PECAM1, and integrins [13].

In this study, the presence of *S. aureus* encoding *selX* on lesional skin was associated with a more severe disease course expressed by objective SCORAD, intensity, dryness, extent of skin lesions and the largest extent of skin lesions during flares in the year preceding the examination. This implies a potent pro-inflammatory effect of SEL-X on AD skin. Furthermore, the unique ability of SEL-X to suppress both innate and acquired immune response combined with its high prevalence suggest that it could be one of the factors responsible for persistent colonization by *S. aureus* in AD. This, as justified by our data, could result in robust dissemination of strains with this SAg and aggravation of the extent and intensity of AD flares. Additionally, the considerable role of SEL-X in causing severe systemic infections should prompt discussion on the recognition of untreated AD individuals as potential populational reservoirs of highly virulent *S. aureus*.

Across different studies, genes encoded on the EGC were frequently isolated in patients with AD [24,28,37]. Two available reports did not show an increased severity of AD in patients colonized by EGC-positive *S. aureus* [24,28]. However, that was also not reported for other enterotoxin genes analyzed in those studies. The vast body of evidence supporting the relation of increased AD severity in patients colonized by SAg-positive *S. aureus,* as well as the evolution of *S. aureus* characterized by increasing prevalence of EGC could therefore suggest that these findings need to be verified [24]. Indeed, the present study showed that carriage of *S. aureus* encoding all genes belonging to the EGC on lesional skin was associated with higher disease severity expressed by SCORAD index and its components. The presence of *seln* correlated with intensity, erythema, and lichenification; *selm* and *selo* with intensity and dryness; and *seg, selI*, and *selU* with lichenification. This underlies the role of EGC cluster in driving disease severity primarily by increasing both the acute (erythema) and chronic (lichenification, dryness) hallmarks of intensity.

Dryness is a clinical sign of a disturbed skin barrier, which occurs both due to intrinsic and extrinsic factors. Therefore, the observed correlation could be explained by the inflammation driven by *S. aureus* EGC products. On the other hand, lichenification was shown to be correlated with activation of Th22 cells [38]. It could therefore be hypothesized that long-standing colonization by EGC-positive S. *aureus* causes the upregulation of Th22 responses and induces the chronicity of AD lesions.

The presence of *seg, selI*, *and selU* on nonlesional skin was observed in patients with a lower extent of skin lesions, while the *selO* and *selM* were identified in patients with higher skin dryness. The presence of *seg, selI* and *selU* in *S. aureus* isolated from the anterior nares corresponded with lower mean values of extent of skin lesions in stable periods of disease, while *selM* and *selO* were identified in patients reporting a less severe itch. The hypothesis regarding the association with skin dryness has already been put forward. The inverse association with the extent of skin lesions on nonlesional skin and the nose, as well as lower itch severity could be explained by the fact that the expression of genes encoding SAgs is regulated by quorum-sensing systems such as agr [39]. These are activated after the density of *S. aureus* reaches a threshold in a given microniche. Therefore, lower abundance of *S. aureus* on the nonlesional skin and in the anterior nares could be responsible for limited expression of those SAgs and lower values of the observed clinical parameters. Furthermore, the subjective, patient-reported character of these characteristics could induce bias and cause the results to be opposite than expected.

In literature, classic staphylococcal SAgs (SAE-SEG, TSST-1) are most frequently reported to play a role in aggravating AD. Most data exist for SEB, which was revealed to stimulate the IL-4, IL-31, and IL-22 responses, and occasionally correlate with disease severity [18,40,41]. Additionally, in contrast to healthy individuals, SEB-stimulated regulatory T cells of AD patients exerted effector Th2-like function [42].

In the present report, carriage of the *sec* gene isolated from the lesional skin was associated with higher mean values of Total SCORAD and DLQI. Nevertheless, *sec* was only detected in 3/43 (7.0%) of strains in this microniche, which suggests a low impact of this SAg on the analyzed population of AD patients. The presence of *sea* gene in strains from the anterior nares was associated with lower mean values of itch, but it was again detected with low prevalence (8.1%). The inverse correlation could result from the aforementioned quorum-sensing-dependent gene expression and be biased by subjective character of this parameter. The low impact of classic SAgs on the course of AD was confirmed in multivariate linear regression significantly predicting dependent clinical variables based only on the presence of *selX* and EGC. The low prevalence and impact of classic SAgs demonstrated in this study seem to confirm a continuous adaptation of *S. aureus* to AD patients, which also probably involves the differential expression of toxins. This is reinforced by a study analyzing temporal differences in SAg gene carriage, in which classic SAg genes were prominently less prevalent after the period of 6–11 years compared with baseline, while members of the EGC were retrieved with similar or higher rates [24].

Many studies reported IgE-specific antibodies against SAgs to be present in the sera of patients with AD. As reported in the recent meta-analysis, the weighted prevalence of IgE against SEA, SEB, and TSST-1 was 33%, 35%, and 16%, respectively. In case of SEA and SEB, the levels of those antibodies were several times higher in patients with AD than in controls [23]. However, the very same presence of SAgs in the patients’ serum could not be correlated to disease severity. Specific IgE were not analyzed in this study. Nevertheless, neither the total number nor the presence of any particular SAg were shown to affect the total serum IgE concentration. This implies that the demonstrated aggravation of AD lesions by *selX* and EGC could be independent of IgE-mediated sensitization.

### 4.1. Therapeutic Implications

The presented results substantiate the hypothesis on the circulation of *S. aureus* encoding various SAgs between the anterior nares and the skin of patients with AD. This seems to cause chronic microbial dysbiosis and to aggravate disease severity. Therefore, investigation of novel treatments reducing the burden of *S. aureus* superantigens in AD should be undertaken. Although antibiotic treatment was shown to be successful in reducing disease severity in patients with superantigen-positive *S. aureus* [19], such a colonization cannot be routinely treated with conventional antimicrobials. Topical agents without the potential of causing bacterial resistance might prove helpful in long-term management of AD. These could implement catechin, apple polyphenols, and 4-hydroxytyrosol, which were shown to inhibit the effects of classic SAgs [43,44,45]. However, their effect on other staphylococcal enterotoxins analyzed in this study has not been assessed, which justifies further studies to elucidate their full clinical potential.

### 4.2. Study Limitations

This study analyzed a wide range of staphylococcal superantigens that had been proposed to affect the course of atopic dermatitis. However, some novel superantigens were not investigated and therefore their role in atopic dermatitis should be studied in the future. Moreover, PCR analysis only allows discovering the presence of genes encoding superantigens but does not provide information on their expression in vivo. Lastly, although the number of patients involved in the analysis was relatively high compared with other reports, some results could have been biased by the sample size.

## 5. Conclusions

This study revealed a high prevalence of *Staphylococcus aureus* encoding superantigens on lesional skin, nonlesional skin, and the anterior nares of adult patients with atopic dermatitis. *selX* was the most prevalent, followed by genes of the enterotoxin gene cluster (*seg, seli, selM, selN, selO, selU*). In contrast to other studies, classic superantigens (*sea, seb, sec, sed, see, tstH)* were isolated with low frequency. The presence of the *selX* and the enterotoxin gene cluster positively correlated with a wide range of clinical parameters reflecting the severity of atopic dermatitis. The high prevalence of *Staphylococcus aureus* and the identified distribution of superantigen genes in the analyzed microniches suggests that the anterior nares and nonlesional skin are significant reservoirs of this pathogen. Presented results justify the investigation of novel treatments to limit the burden of staphylococcal superantigens in atopic dermatitis.

## Figures and Tables

**Figure 1 cells-11-03921-f001:**
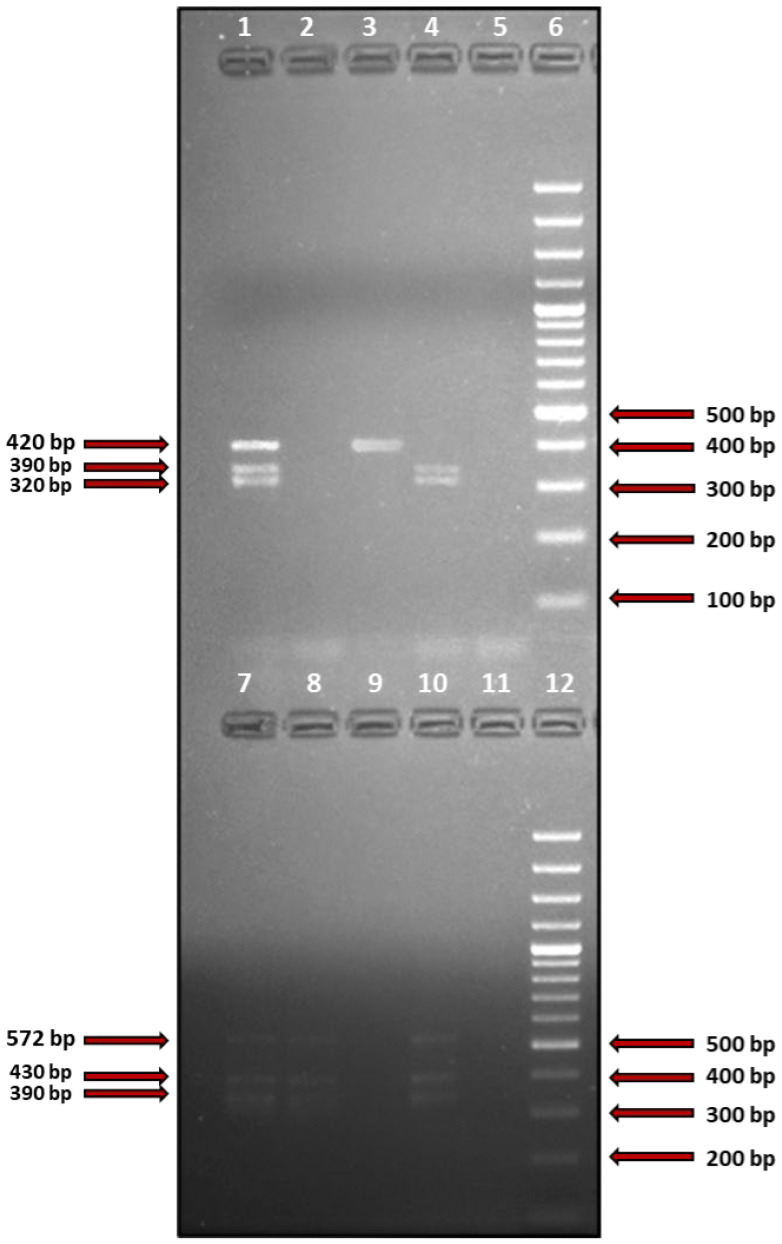
Visualization of the products of multiplex-PCR reaction performed on DNA extracted from exemplary clinical isolates of *Staphylococcus aureus*. The reaction was positive for *selM, selO,* and *selX* (lanes 1–5, amplicon sizes: 320 bp, 390 bp, and 420 bp, respectively); seg, selI, and selU (lanes 7–12, amplicon sizes: 430 bp, 572 bp, and 390 bp, respectively). Lanes 11 and 22 contain a DNA molecular-weight size marker (100 bp DNA Ladder, Thermo Fisher Scientific, Bartlett, IL, USA).

**Figure 2 cells-11-03921-f002:**
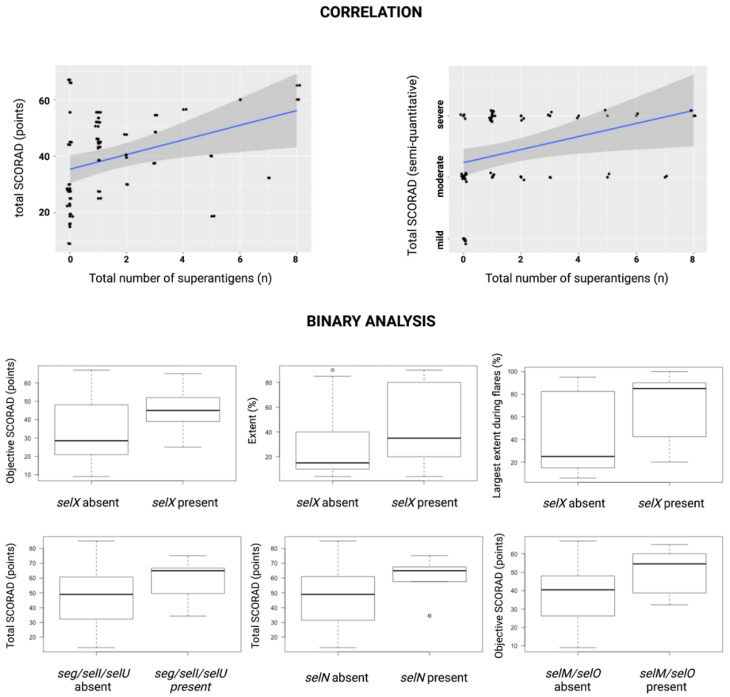
Graphs depicting the most significant results of the statistical analysis. The upper part shows Spearman rank correlations between the total number of superantigens identified on lesional skin and total SCORAD/semi-quantitative assessment of total SCORAD. The lower part demonstrates statistically significant differences in extent, objective SCORAD, largest extent of skin lesions during the flares during the year preceding the examination, and total SCORAD based on the presence or absence of *selX* and superantigens encoded on the enterotoxin gene cluster (*seg/selI/selU, selM/selO, selN*).

**Table 1 cells-11-03921-t001:** Characteristics of the primers used in this study.

Name of the Gene	Name of the Primer	Nucleotide Sequence of the Primer (5′→3′)	PCR Product Size (bp)	Reference
*sea*	SEA-F	GATTCACAAAGGATATTGTTGATAAATAT	400	Salgado-Pabon W, et al.
	SEA-R	GTCCTTGAGCACCAAATAAATC		
*seb*	SEB-F	GTATGATGATAATCATGTATCAGCAA	625	Salgado-Pabon W, et al.
	SEB-R	CGTAAGATAAACTTCAATCTTCACAT		
*sec*	SEC-F	GAGTCAACCAGACCCTATGCC	650	Salgado-Pabon W, et al.
	SEC-R	CGCCTGGTGCAGGCATC		
*sed*	SED-F	GCATTACTCTTTTTTACTAGTTTGGTA	530	Salgado-Pabon W, et al.
	SED-R	CCTTGCTTGTGCATCTAATTC		
*see*	SEE-F	CTGAATTACAAAGAAATGCTTTAAGC	420	Salgado-Pabon W, et al.
	SEE-R	GCCTTGCCTGAAGATCTA		
*seg*	SEG-F	GGGAACTATGGGTAATGTAATGAATC	430	Vu BG, et al.
	SEG-R	TGAGCCAGTGTCTTGCTTTG		
*sel*I	SEL-I-F	GCTCAAGGTGATATTGGTGTAGG	572	Vu BG, et al.
	SEL-I-R	CTTACAGGCAGTCCATCTCC		
*sel*M	SEL-M-F	CGGTGGAGTTACATTAGCAGGT	320	Vu BG, et al.
	SEL-M-R	TTTCAGCTTGTCCTGTTCCA		
*sel*N	SEL-N-F	GCTTATACGGAGGAGTTACG	298	Vu BG, et al.
	SEL-N-R	GCTCCCACTGAACCTTTTACG		
*sel*O	SEL-O-F	GGAATTTAGCTCATCAGCGATT	390	Vu BG, et al.
	SEL-O-R	TGCTCCGAATGAGAATGAAA		
*sel*U	SEL-U-F	GCAGCTTACTATTTATGTTAAATGGC	390	Vu BG, et al.
	SEL-U-R	CTATTTGATTTCCATCATGCTCGG		
*sel*X	SEL-X-F	TCTATGGGGGAACATTTGGA	420	Salgado-Pabon W, et al.
	SEL-X-R	CCGCCATCTTTTGTATTTATGA		
*tst*H	TSST-1-F	GAAATTTTTCATCGTAAGCCCTTTGTTG	655	Salgado-Pabon W, et al.
	TSST-1-R	TTCATCAATATTTATAGGTGGTTTTTCA		

**Table 2 cells-11-03921-t002:** Detailed characteristics of the individuals with atopic dermatitis involved in the study.

Females/males	17/33 (34%/66%)
Age	range 18–70, mean 32.7 ± 12.9
SCORAD index (points)	range 13–85, mean 49.1 ± 17.2
<25 points (mild AD)	5/50 (10%)
25–50 points (moderate AD)	19/50 (38%)
>50 points (severe)	26/50 (52%)
Extent—% of involved body area	range 4–90, mean 37.6 ± 32
Intensity (points):	
Redness	range 0–3, mean 1.62 ± 0.78
Swelling	range 0–3, mean 1.4 ± 0.67
Oozing/crust	range 0–3, mean 1.08 ± 0.99
Scratch marks	range 0–3, mean 1.54 ± 0.84
Lichenification	range 0–3, mean 1.92 ± 0.78
Dryness	range 0–3, mean 1.52 ± 0.93
Subjective symptoms (points):	
Itch	range 0–10, mean 6.0 ± 2.77
Sleeplessness	range 0–10, mean 3.84 ± 3.59
Objective SCORAD (points)	range 9–67, mean 39.3 ± 15.1
Largest extent of skin lesions during flares in the year preceding the examination (%)	range 6–100, mean 56.2 ± 34.6
Extent of skin lesions during stable periods of the disease (%)	range 0–15, mean 4.68 ± 3.68
Total IgE (UI/mL)	range 5.4–2500, mean 1354 ± 1081

**Table 3 cells-11-03921-t003:** Genes encoding staphylococcal superantigens detected in the strains isolated from the lesional skin, nonlesional skin, and the anterior nares of the study group.

Gene	Lesional Skin, n (%)	Nonlesional Skin, n (%)	Anterior Nares, n (%)
*sea*	3 (7.0%)	2 (6.1%)	3 (8.1%)
*seb*	1 (2.3%)	-	-
*sec*	3 (7.0%)	1 (3.0%)	2 (5.4%)
*sed*	1 (2.3%)	-	2 (5.4%)
*see*	-	-	-
*seg*	7 (16.3%)	9 (27.3%)	3 (8.1%)
*selI*	7 (16.3%)	9 (27.3%)	3 (8.1%)
*selM*	7 (16.3%)	9 (27.3%)	5 (13.5%)
*selO*	7 (16.3%)	9 (27.3%)	5 (13.5%)
*selN*	9 (20.9%)	10 (30.3%)	9 (24.3%)
*selU*	7 (16.3%)	9 (27.3%)	3 (8.1%)
*selX*	23 (53.5%)	22 (66.7%)	15 (40.5%)
*tstH*	1 (2.3%)	2 (6.1%)	2 (5.4%)

## Data Availability

Source data are available in an online data repository: https://data.mendeley.com/datasets/5zjxtpc547/1 (accessed on 30 September 2022).

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
