# Peer review of "Enterotoxin Gene Cluster and selX Are Associated with Atopic Dermatitis Severity—A Cross-Sectional Molecular Study of Staphylococcus aureus Superantigens"

_cells, 2022, doi:10.3390/cells11233921_

Round 1
Reviewer 1 Report
Thanks you for giving me the opportunity to review this paper. In the manuscript, the authors isolate S. aureus from three different locations (lesional skin, non-lesional skin, nares) on the skin of patients with atopic dermatitis. Subsequently, they determine the presence of different superantigens and evaluate the association of lesional superantigens with disease severity.
I have only a few questions and comments:
-The authors present many correlation analyses, but the intention is sometimes not entirely clear to me t first sight. Maybe you can include a few sentences on the rationale for these analyses in the respective parts of the results section? (E.g. to analyze whether xy could be a reservoir for Z...)
- I'm not sure I understand the sentence starting in line 310. What does the statistical significance refer to?
- Whereas the authors found the smallest amount of S. aureus on non-lesional skin and a relatively high amount in the nares, the number of superantigen genes identified is much higher on non-lesional skin than in the nares or even on lesional skin. Is this chance finding, or can you envisage some kind of biological rationale for this?
- I think figure 1 is rather large and could be limited to the relevant areas of the gel.
- Can you speculate on the reasons why you didn't identify the classical superantigens very often compared to other studies, and they didn't correlate with disease severity? Also, do you have any idea why none of the other studies identified the correlation of EGC with disease severity?
Author Response
Dear Reviewer,
Thank you for revising our manuscript regarding the role of staphylococcal superantigens in atopic dermatitis and for proposing valuable corrections that will improve the academic value of the article. We have made the suggested changes in the manuscript. Below we enclose point-by-point response to the review:
-The authors present many correlation analyses, but the intention is sometimes not entirely clear to me t first sight. Maybe you can include a few sentences on the rationale for these analyses in the respective parts of the results section? (E.g. to analyze whether xy could be a reservoir for Z...)
We thank the Reviewer for the valuable remark improving the clarity of the manuscript. In the results section, we included several sentences explaining the rationale of the analyses (please see p. 5, lines 183-185; p. 8, lines 287-289; p. 8, lines 317-318)
- I'm not sure I understand the sentence starting in line 310. What does the statistical significance refer to?
We thank the Reviewer for pointing the lack of clarity in the sentence. We modified it to underline that the statistical significance referred to the multiple regression model considering all of the listed variables (currently p. 9, line 340)
- Whereas the authors found the smallest amount of S. aureus on non-lesional skin and a relatively high amount in the nares, the number of superantigen genes identified is much higher on non-lesional skin than in the nares or even on lesional skin. Is this chance finding, or can you envisage some kind of biological rationale for this?
We thank the Reviewer for pointing out the lack of proper discussion of the presented results. We updated the manuscript to provide possible explanation of the highest number of identified superantigens on the nonlesional skin (p. 10, lines 381-387).
- I think figure 1 is rather large and could be limited to the relevant areas of the gel.
We thank the Reviewer for the suggestion regarding the graphic content of the manuscript. We updated the Figure according to the suggestions (currently present on p. 6).
- Can you speculate on the reasons why you didn't identify the classical superantigens very often compared to other studies, and they didn't correlate with disease severity? Also, do you have any idea why none of the other studies identified the correlation of EGC with disease severity?
We thank the Reviewer for pointing out the lacking parts of discussion. We updated the manuscript with the missing content (please refer to p. 11, lines 445-450; p. 12, lines 496-501).
Reviewer 2 Report
Good and interesting paper. Congrats!
Author Response
Good and interesting paper. Congrats!
Dear Reviewer,
Thank you for revising our manuscript regarding the staphylococcal enterotoxins in atopic dermatitis and for recommending it for acceptation in the Journal.